# AutoMixQ: Automatic Mixed-precision Quantization for Deploying Bit-Efficient LLMs

## Abstract

Quantization has become a critical technique for efficiently deploying large language models (LLMs), as their massive size makes full-precision inference impractical on most hardware. Among various quantization strategies, 4-bit post-training quantization (PTQ) strikes a favorable balance between compression and performance for hardware-accelerated deployment. Further reducing precision below 4 bit would further increase efficiency, but often leads to severe performance degradation. This dilemma stems from two overlooked issues: 1) most PTQ methods primarily focus on reducing quantization error caused by salient channels with larger magnitude, while neglecting the importance of unremarkable channels with lower magnitude but high semantic relevance; 2) most PTQ methods typically apply quantization based on layer-wise reconstruction loss, failing to account for the cumulative and interdependent effects across layers. In this work, we present AutoMixQ, an automatic mixed-precision quantization to address the two key challenges of sub-4-bit quantization. Instead of focusing solely on salient channels, AutoMixQ considers both salient and unremarkable channels by introducing mixed-bit strategies that capture diverse quantization sensitivities across channels. AutoMixQ first constrains the search space derived from prior empirical observations for stability and efficiency. Then, it conducts an automatic search guided by distillation and global losses that model intra- and inter-layer dependencies, achieving holistic optimized performance. Experiments on several LLMs demonstrate that AutoMixQ achieves better accuracy under low-bit settings, outperforming existing methods by producing more balanced and effective bit allocation.

## 1 Introduction

Large language models (LLMs) (Bubeck et al., 2023; OpenAI, 2025; DeepSeek-AI et al., 2025) have achieved impressive success across a wide range of tasks, driven by ever-increasing scales of model parameters, training data, and computation. This scaling trend has led to substantial gains in reasoning and generalization, but also brings significant resource demands. For instance, DeepSeek-R1 features 671 billion total parameters and 37 billion activated parameters, requiring over 1300 GB of memory in FP16 precision—necessitating at least 14 NVIDIA H100 GPUs for inference. As model sizes continue to grow, optimizing inference efficiency becomes increasingly important, where LLM quantization has become a widely adopted technique for reducing memory consumption and computational cost by compressing weights and activations into low-bit representations.

Building on this motivation, quantization techniques for LLMs generally fall into two categories: quantization-aware training (QAT) (Jacob et al., 2018; Liu et al., 2023) and post-training quantization (PTQ) (Frantar et al., 2022; Lin et al., 2024; Dettmers et al., 2022). While QAT integrates quantization into training and achieves good accuracy at low bit-widths, it is often impractical for large-scale models due to the need for full retraining and labeled data. In contrast, PTQ calibrates quantization parameters using a small unlabeled dataset without updating model weights, making it far more scalable and appealing for LLM deployment. Among PTQ methods, 4-bit quantization has become a widely accepted standard, offering a practical balance between compression ratio and model accuracy (Dettmers & Zettlemoyer, 2023). Pushing precision further below 4 bits promises even greater efficiency (Ma et al., 2024; Daniel Han & team, 2025), but existing PTQ methods encounter limitations that hinder effective quantization in these regimes. In particular, 3-bit quan-

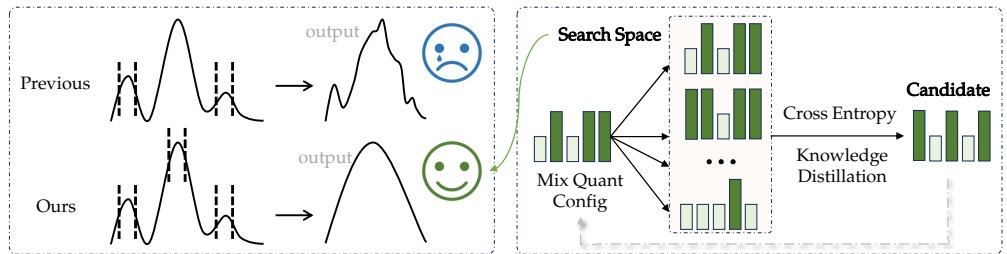

Figure 1: Overview of **AutoMixQ**, an automatic mixed-precision quantization framework for sub-4-bit LLM deployment. It combines mixed-precision protection, which preserves both large- and small-magnitude channels based on sensitivity, with an automatic global search that allocates bits across layers by modeling intra- and inter-layer dependencies. These strategies yield globally balanced precision assignments.

tization lacks hardware acceleration, motivating practical strategies that mix only 2-bit and 4-bit precision, which is a direction that remains underexplored.

These limitations can be summarized into two aspects. **Channel-level oversight**: most approaches (Dettmers et al., 2022; Lin et al., 2024; Xiao et al., 2023) focus heavily on reducing quantization error caused by salient channels—those with large magnitudes or outlier statistics—while neglecting unremarkable channels that exhibit lower amplitudes but carry high semantic relevance for model outputs (Dettmers et al., 2022; Lee et al., 2024a). These subtle yet important channels are often quantized indiscriminately (Li et al., 2023; Lee et al., 2024a), resulting in degraded expressiveness. **Layer-level myopia**: existing PTQ methods (Frantar et al., 2022; Lin et al., 2024) typically adopt a layer-wise quantization scheme, independently optimizing each layer based on local reconstruction loss. Such greedy strategies fail to consider cumulative errors and inter-layer dependencies, often leading to locally optimal but globally suboptimal under aggressive low-bit regimes.

In this paper, we introduce **AutoMixQ**, as shown in Figure 1, an automatic mixed-precision quantization framework designed for sub-4-bit LLM deployment. AutoMixQ is built upon two key strategies. **Mixed-precision protection** explicitly considers both salient and unremarkable channels: instead of protecting only large-magnitude outliers, it assigns higher widths to channels with the largest and smallest absolute magnitudes based on their quantization sensitivity, ensuring that semantically important but low-magnitude channels are preserved. **Automatic global search** further optimizes bit allocation across layers by modeling intra- and inter-layer dependencies. Guided by distillation and cross-entropy losses from a global view, this search yields holistically balanced precision assignments rather than layer-wise local optima. Extensive experiments on several LLMs demonstrate that AutoMixQ consistently outperforms prior PTQ approaches at sub-4-bit settings, achieving comparable or superior accuracy while enabling more efficient and practical deployment.

The main contributions are summarized as follows.

- We conduct a systematic analysis of sub-4-bit PTQ that uncovers the oversight of semantically important low-magnitude channels and the layer-wise myopia in local optimization strategies.
- AutoMixQ framework combines mixed-precision protection to preserve expressiveness with automatic global search to achieve consistent, layer-aware precision allocation.
- We conduct extensive experiments on representative LLMs, showing that AutoMixQ achieves accuracy–efficiency trade-offs in sub-4-bit regimes, outperforming prior PTQ approaches.

## 2 RELATED WORK

Mixed-precision quantization with search techniques has long served as a fundamental technique for compressing deep models. Some researches successfully automatically search a high-expressiveness deep neutal network architecture for IoT devices (Sun et al., 2022). Besides, mixed-precision quantization search techniques have also been explored on some pre-trained language models such as BERT(Zhao et al., 2021). With the emergence of LLMs, these classic methods—such as QAT (Jacob et al., 2018; Liu et al., 2023; 2025) and PTQ (Frantar et al., 2022; Lin et al., 2024; Dettmers

et al., 2022)—have been reintroduced to address new challenges in scale and accuracy. Although the core strategies remain largely unchanged, adapting them to the unique characteristics of LLMs has sparked a new wave of specialized solutions. These include training-aware methods tailored for data scarcity and sensitivity, as well as search-based PTQ approaches that optimize quantization configurations.

**Quantization-Aware Training.** LLM-QAT (Liu et al., 2023) is the first to explore QAT for LLMs, using data-free knowledge distillation to align full-precision teacher logits with quantized student logits. BitDistiller (Du et al., 2024) enables asymmetric quantization via an asymmetric clipping strategy during self-distillation. EfficientQAT (Chen et al., 2024) reduces overhead by splitting QAT into two phases: first optimizing all parameters within each block, then refining quantization parameters globally. BitNet (Wang et al., 2023) replaces standard Linears with BitLinears and trains from scratch, while its variant BitNet b1.58 (Ma et al., 2024) achieves near-lossless performance using ternary weights. ParetoQ (Liu et al., 2025) further improves training schemes and quantization functions, achieving strong performance across diverse settings.

**Post-Training Quantization.** PTQ is a training-free quantization technique widely adopted for its simplicity and effectiveness. However, applying PTQ to LLMs is challenging due to their scale and structural complexity. In this work, we focus on weight-only PTQ methods. GPTQ (Frantar et al., 2022), built on the OBQ (Frantar & Alistarh, 2022) framework, introduces layer-wise optimization with Hessian-guided refinements and dynamic weight updates to reduce reconstruction loss. Many approaches target outliers in LLMs: LLM.int8() (Dettmers et al., 2022) uses empirical heuristics to detect outlier features, while AWQ (Lin et al., 2024) rescales weights based on activation magnitudes to mitigate mean square error. OmniQuant (Shao et al., 2023) proposes Learnable Weight Clipping and Learnable Equivalent Transformation to improve quantization robustness. Other methods like SpQR (Dettmers et al., 2023), SqueezeLLM (Kim et al., 2024), and OWQ (Lee et al., 2024a) aim to isolate and preserve outlier weights. QuiP (Chee et al., 2023) addresses ultra-low-bit settings via incoherence processing, while ZeroQuant (4+2) (Wu et al., 2023) leverages FP6 quantization to improve precision.

## 3 METHODOLOGY

In this section, we first review the preliminary of Quantization on LLMs. Then, to address the issue of **channel-level oversight**, we introduce a **mixed-precision protection**. Next, to overcome **layer-level myopia**, we further propose an **automatic global search** strategy for balanced bit allocation across layers. Together, these components present the components of the proposed **AutoMixQ** framework, with the overall pipeline illustrated in Figure 1.

### 3.1 QUANTIZATION PRELIMINARY

Quantization is a widely used technique to compress neural networks (Gholami et al., 2022; Guo, 2018; Zhang et al., 2018) by reducing parameter precision from high bit-width (e.g., FP16/FP32) to lower bit-width integers. This reduces both memory footprint and computational cost, making deployment of large models more efficient. Formally, given a tensor $\mathbf{X}$ with range $[x_{\min}, x_{\max}]$ and target bit-width $N$, the quantized value is

$$\mathbf{X}_q = \text{clamp}\Big( \Big\lfloor \mathbf{X}_q/s + z \Big\rceil, \ -2^{N-1}, \ 2^{N-1} - 1 \Big), \quad s = (x_{\max} - x_{\min})/2^N - 1, \quad z = -\Big\lfloor x_{\min}/s \Big\rceil, \ (1)$$

where $\lfloor \cdot \rceil$ represents nearest rounding, $s$ is the quantization scale that maps the floating-point range to $2^N$ discrete levels, and $z$ is the integer zero-point used to align the minimum value $x_{\min}$. After quantization, the original values are approximately restored by dequantization, which maps integers back to floating-point space. Formally, the dequantized value is

$$\mathbf{X}' = (\mathbf{X}_q - z) \cdot s, \tag{2}$$

The discrepancy between $\mathbf{X}'$ and the original tensor $\mathbf{X}$ constitutes the quantization error, which accumulates across layers in large models.

As the scale of LLMs grows, quantization faces new challenges due to the massive parameter count and long sequence lengths. In practice, quantization is predominantly applied to the projection linear

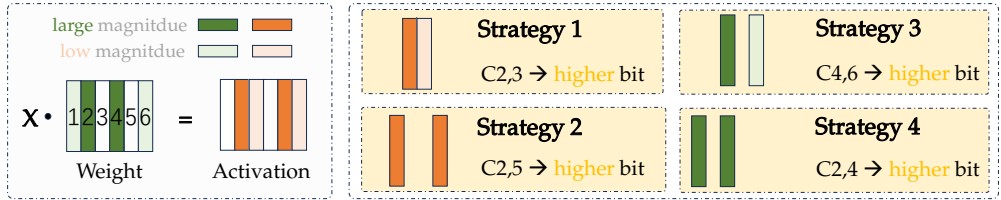

Figure 2: Four mixed-precision protection strategies in AutoMixQ. Strategies 1–2 assign higher bit-widths based on output activations (preserving both top and low values or only top), while Strategies 3–4 follow the same logic using weight magnitudes.

layers in attention and feed-forward modules (Dettmers et al., 2022). Formally, a linear layer can be written as

$$\mathbf{Y} = \mathbf{X}\mathbf{W}, \tag{3}$$

where $X$ denotes the activation, $W$ the weight matrix, and $Y$ the output. The typical quantization objective is to minimize the reconstruction error between full-precision and quantized outputs:

$$\min_{\mathcal{Q}(\cdot)} \ \|\mathbf{X}\mathbf{W} - \mathbf{X}\mathcal{Q}(\mathbf{W})\|, \tag{4}$$

where $\mathcal{Q}(\cdot)$ is the quantization operator applied to the weights. Prior studies have demonstrated that a small fraction of extreme outlier channels disproportionately contribute to this error (Zheng et al., 2024; Dettmers et al., 2022), motivating mixed-precision strategies that preserve critical channels.

### 3.2 MIXED-PRECISION PROTECTION

**Channel-level Oversight.** In prior studies (Dettmers et al., 2023; Kim et al., 2024; Lee et al., 2024a), salient weight channels or outlier activations have received particular attention. Since their magnitudes are significantly larger than others, quantizing them with ultra-low-bit precision often leads to severe information loss and unacceptable performance degradation (Li et al., 2023). To mitigate this, these outliers are usually assigned higher bit-widths, allowing for more precise representation. While effective when the precision is above 4 bits, this strategy fails to sustain performance when quantization drops below 4 bits—often resulting in sharp accuracy degradation.

We revisit this problem and attribute the failure of existing mixed-precision quantization approaches to an implicit assumption: that protecting only large-magnitude channels—either in weights or activations—is sufficient to preserve the final output distribution. However, as illustrated in Figure **??**, some subtle yet important channels also have a non-negligible impact on the output, despite their low individual magnitudes. Focusing solely on outliers can reduce local quantization error, but fails to guarantee the consistency of the final model behavior, especially under aggressive quantization.

**Quantization Protection Mechanism.** Motivated by this insight, we propose a flexible protection mechanism that identifies important channels based on the statistical distributions of both weights and output activations. As shown in Figure 2, each weight channel can be assigned to one of four different mixed-precision protection strategies:

- **Strategy 1**: Assigns higher bit-widths to channels corresponding to both top and low output activations; the remaining channels are quantized with lower bit-widths.
- **Strategy 2**: Allocates higher precision only to channels associated with the top output activations.
- **Strategy 3**: Applies Strategy 1 based on the magnitude of weight channels instead of activations.
- **Strategy 4**: Applies Strategy 2 based on the weight magnitude.

All channel selections are made based on statistics computed from a small held-out calibration set, to avoid overfitting to any specific downstream domain. On the one hand, this design enables grouping 4-bit and 2-bit channels for practical acceleration from an engineering perspective. On the other hand, we argue that weights should not be constrained to a single mixed-precision heuristic. Compared to heuristic approaches that focus only on salient or extreme channels, our method provides greater flexibility in defining importance per weight—leading to improved accuracy preservation and robustness. These advantages will be further substantiated in the experimental section.

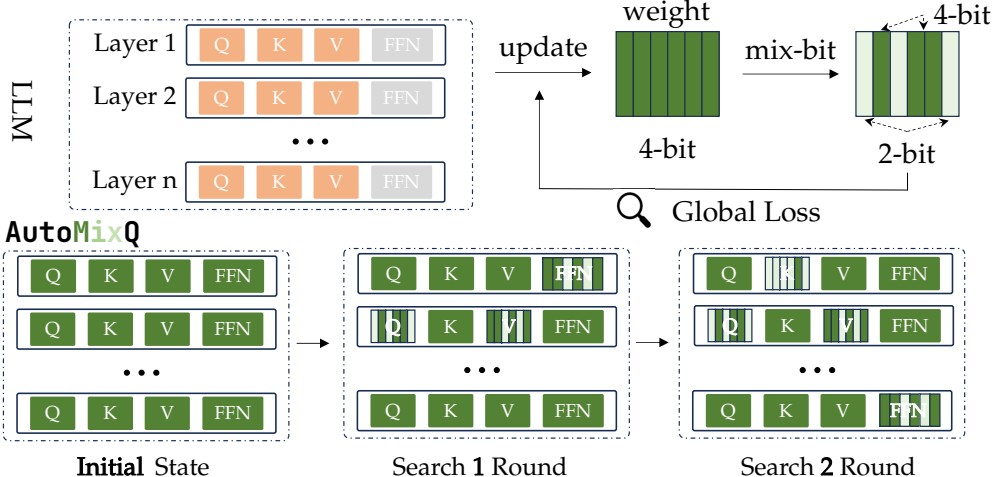

Figure 3: The pipeline of our AutoMixQ: The framework begins with sensitivity analysis to identify candidates for mixed-precision quantization, followed by iterative global search with perturbation-based sampling and performance evaluation. A task-aware loss combining cross-entropy and knowledge distillation guides the update of bit allocations toward a globally optimal configuration.

## 3.3 Automatic Global Search

**Layer-level Myopia.** A growing body of mixed-precision quantization research (Yu et al., 2024; Li et al., 2023; Dettmers et al., 2023; Kim et al., 2024) has recognized that different weights exhibit varying sensitivities to quantization noise. Many existing methods estimate this sensitivity using approximations such as first-order gradients or second-order Hessian traces (Zheng et al., 2024), and heuristically assign higher bit-widths to sensitive layers to reduce output distortion. However, these approaches typically adopt a layer-wise quantization scheme, where each layer is quantized independently without considering the propagation of quantization error through the network. For instance, methods like GPTQ and AWQ perform greedy, local optimization at the layer level, neglecting the cumulative and interdependent effects of quantizing earlier layers on subsequent ones. We refer to this limitation as **layer-level myopia**, which often leads to locally optimal yet globally suboptimal precision configurations. This motivates the need for a global, model-aware precision allocation strategy that jointly optimizes all layers in a coordinated manner.

**Automatic Global Search.** To overcome this limitation, we introduce **AutoMixQ**, an automatic search framework for discovering globally optimal mixed-precision configurations under low-bit constraints. The full search pipeline is summarized in Algorithm 1, and illustrated in Figure 3. The procedure begins with a mixed-precision sensitivity analysis across all weights, identifying channels that are suitable for sub-4-bit quantization. Based on these statistics, a candidate set of weights is selected to construct an initial search space through random perturbations. Each sample in the search space corresponds to a particular per-layer bit-width assignment and is evaluated using a quantized model.

After evaluating the samples, the best-performing configuration is used to update the precision assignment, while other high-performing candidates are retained to generate new samples via perturbation. This process is iteratively repeated until convergence. To guide the search process, we define a task-specific loss function:

$$\mathcal{L} = \mathcal{L}_{\text{CE}} + \alpha \cdot \mathcal{L}_{\text{KD}}, \tag{5}$$

where $\alpha$ balances the contributions of the task and distillation objectives. Specifically, given the predicted logits $\hat{y}$ from the quantized model, the true label $y$, and the original model logits $\overline{y}$, the two components are:

$$\mathcal{L}_{\text{CE}} = \text{CrossEntropy}(y, \hat{y}),$$

$$\mathcal{L}_{\text{KD}} = T^2 \cdot \text{KL}\left(\text{softmax}\left(\frac{\overline{y}}{T}\right) \,\middle\|\, \text{softmax}\left(\frac{y}{T}\right)\right),$$

where $T$ is the temperature parameter. This loss encourages the quantized model to match both the ground-truth labels and the soft distribution of the full-precision model. The global search is thus

---

**Algorithm 1:** The procedure of automatic search mixed-precision quantization (AutoMixQ)

---

**Input:** Candidate set $\mathcal{S}_{can}$, search space size $N$, elite size $E$, sampling step $n$, mixde-precision quantization starting configuration $q$, cycle condition $C$.

**Output:** The optimal mixde-precision quantization configuration $q_{out}$.

**1 while** $q$ *meet* $C$ **do**

**2**     Generate the search space $M$ from $\mathcal{S}_{can}$. $M$ contains $N$ samples, and each sample contains $n$ different individual weights to be mixed-precision quantized.

**3**     $Record \leftarrow \{\}$

**4**     **for** $i = 1, 2, \cdots, N$ **do**

**5**        Select the i-th sample $m_i$ from $M$, update $q$ with $m_i$, and obtain a new quantization configuration $q_{tmp}^i \leftarrow q + q_{top}^i$

**6**        Apply $q_{tmp}^i$ to the target model, and record the final loss $L_{tmp}^i$

**7**        $Record.update(\{L_{tmp}^i : q_{tmp}^i\})$

**8**     Gather sorted quantization configuration list $Q$ by $L_{tmp}^i$, update quantization configuration with the top sample $q \leftarrow Q.pop()$.

**9**     Obtain a new $\mathcal{S}_{can}$ with the remaining samples in order. Speciallly,

**10**     $\mathcal{S}_{can} \leftarrow []$

**11**     **while** $len(\mathcal{S}_{can}) < E$ **do**

**12**        $\mathcal{S}_{can}.append(Q.pop())$

---

driven by minimizing this composite loss, ensuring task accuracy and semantic consistency during quantization.

# 4 EXPERIMENT

## 4.1 IMPLEMENTATION DETAILS

Our study primarily investigates sub-4-bit mixed-precision quantization, as these settings can effectively preserve the performance of LLMs under aggressive compression (Dettmers & Zettlemoyer, 2023). For AutoMixQ, we concentrate exclusively on 4-bit and 2-bit mixed-precision quantization, as this regime is practically promising for real acceleration with prospective hardware support. Unless otherwise specified, all reported results are therefore based on 2/4bit mixed-precision quantization. For completeness, since 3-bit precision has also gained increasing acceptance in recent work, we include detailed results for 2/3/4-bit mixed-precision quantization in the Appendix.

**Paramerters setting.** AutoMixQ essentially is a searching framework with the primary objective of obtaining a globally optimal bit allocation scheme. Therefore, it is compatible with fundamental quantization methods such as AWQ (Lin et al., 2024) or GPTQ (Frantar et al., 2022). In this work, all experiments were implemented with a default configuration of group size 128. For the search loss function in Section 3.3, we set $\alpha = 1$ and $T = 2$, respectively. To enable efficient search, we modify the hyperparameters such as $\mathcal{S}_{can}, N$ and $n$ in global search according to the model size. We conduct all the experiments on NVIDIA A100 (80G) GPUs.

**Baselines.** We compare AutoMixQ against two widely adopted quantization methods, GPTQ and AWQ, across multiple tasks. The commonly used vanilla round-to-nearest quantization (RTN) evidently performs poorly under low-bit settings and is therefore excluded from our study. Instead, we conduct an additional layer-wise search based on AWQ for a mixed-precision configuration, since AWQ usually performs better than GPTQ (Lee et al., 2024b) and is typically implemented at the layer level.

**Models and Datasets.** AutoMixQ leverages a small calibration set from the Pile dataset (Gao et al., 2020) for channel-wise numerical statistics, and the global search is performed using the validation set of WikiText-2 (Merity et al., 2017) as the auxiliary dataset. AutoMixQ and the baseline are evaluated on representative and widely adopted models, including Llama3-8B (Meta., 2024), Llama2-13B (Touvron et al., 2023), Qwen2.5-7B, and Qwen2.5-14B (Yang et al., 2024).

Table 1: Perplexity evaluation on wikitext2, where lower values (↓) indicate better language modeling quality under quantization.

|  |  | Llama | | Qwen | |
|---|---|---|---|---|---|
|  |  | 2-13B | 3-8B | 2.5-7B | 2.5-14B |
| GPTQ | W2A16 | 28.10 | 3.53e3 | 78.47 | 46.41 |
|  | W4A16 | 5.00 | 7.18 | 7.11 | 5.72 |
| AWQ | W2A16 | 1.22e5 | 1.55e6 | 1.05e7 | 3.62e7 |
|  | W4A16 | 4.97 | 6.64 | 7.09 | 5.67 |
|  | W3.25A16 | 1.14e5 | 1.25e6 | 9.15e4 | 2.29e7 |
|  | W3.5A16 | 9.93e4 | 3.48e4 | 8.68 | 2.02e7 |
|  | W3.75A16 | 5.37 | 7.77 | 7.64 | 6.19 |
| AutoMixQ | W3.0A16 | 6.62 | 11.26 | 9.65 | 7.92 |
|  | W3.25A16 | 5.60 | 9.31 | 8.58 | 7.25 |
|  | W3.5A16 | 5.27 | 8.11 | 7.98 | 6.65 |
|  | W3.75A16 | 5.09 | 7.14 | 7.49 | 6.12 |

**Tasks and Metrics.** For comprehensive evaluation, we first report the perplexity (PPL) performance on the split test of WikiText-2 dataset (Merity et al., 2017) over AutoMixQ and the baselines, which provides a standard measure of language modeling quality under quantization. In addition, we assess the models on a suite of widely used zero-shot benchmarks, including MMLU (Hendrycks et al., 2020), HellaSwag (Zellers et al., 2019), Winogrande (Sakaguchi et al., 2021), PiQA (Bisk et al., 2020), ARC-E, and ARC-C (Clark et al., 2018) on the widely used unified framework Im-eval(Gao et al., 2024). Evaluating across this diverse set of tasks enables us to capture the comprehensive performance of the quantized models, thereby providing a more reliable reflection of the effectiveness of different quantization strategies.

## 4.2 Downstream Tasks Evaluation

To examine the language modeling quality under quantization, we design an experiment comparing AutoMixQ with baselines on WikiText across different models, as shown in Table 1. For fairness, the baselines are also evaluated under random mixed-precision quantization, with findings shown as below:

- Quantization down to 2 bits poses a severe challenge for conventional methods. Even when restricted to a small fraction of parameters (e.g., 20% of layers, resulting in an average precision of 3.5 bits), the degradation for most models remains catastrophic.

- AutoMixQ can allocate bit-widths to model outputs more rationally by accurately identifying output-insensitive channels and assigning them lower precision. As the proportion of 2-bit allocations increases, it ensures a smooth and stable degradation in accuracy, thereby maintaining consistently acceptable performance at the same average bit-width.

To evaluate model performance under different quantization strategies, we conduct experiments on Llama models across several widely used zero-shot tasks, as shown in Table 2. For fairness, we include baselines with random mixed-precision quantization, and we further report 3-bit baselines as an additional reference, since 3-bit quantization can still deliver acceptable performance. We observe that:

- With only 2-bit and 4-bit quantization at the same average bit-width, AutoMixQ achieves a favorable accuracy–efficiency trade-off, consistently outperforming the baseline across both individual tasks and overall performance.

- AutoMixQ attains an average bit-width of about 3.5 comparable to the 3-bit baseline. Although slightly higher, this performance is achieved using only a mixture of 2-bit and 4-bit quantization, which is theoretically more amendable to hardware acceleration. The resulting accuracy remains within an acceptable range.

Table 2: The accuracy of various zero-shot tasks, where higher values (↑) indicate better language modeling quality under quantization..

|  |  | MMLU | Hella. | PiQA | ARC_E | ARC_C | Wino. | Avg. |
|---|---|---|---|---|---|---|---|---|
| Llama2-13B | AWQ W3A16* | 48.67 | 58.47 | 77.15 | 77.82 | 45.82 | 72.61 | 63.42 |
|  | AWQ W3.5A16 | 26.89 | 28.76 | 56.75 | 31.23 | 23.21 | 60.3 | 37.86 |
|  | AutoMixQ W3.5A16 | **48.08** | **58.25** | **78.02** | **77.23** | **46.76** | **72.61** | **63.49** |
|  | AWQ W3.75A16 | 49.52 | 58.97 | 78.4 | 78.49 | 45.65 | 70.8 | 63.64 |
|  | AutoMixQ W3.75A16 | **50.03** | **59.07** | **78.67** | **78.62** | **46.50** | **71.82** | **64.12** |
| Llam3-8B | AWQ W3A16* | 53.72 | 54.85 | 78.51 | 76.22 | 43.00 | 70.48 | 62.80 |
|  | AWQ W3.5A16 | 42.90 | 40.96 | 67.85 | 62.21 | 37.03 | 65.67 | 52.77 |
|  | AutoMixQ W3.5A16 | **55.45** | **55.48** | **77.69** | **77.86** | **45.73** | **69.38** | **63.60** |
|  | AWQ W3.75A16 | 54.06 | 56.65 | 78.18 | 78.54 | **47.10** | 70.40 | 64.16 |
|  | AutoMixQ W3.75A16 | **57.91** | **58.07** | **79.05** | **79.17** | 46.67 | **70.96** | **65.31** |

* In this experiment, all weights are uniformly quantized to 3-bit instead of mixed-precision.

### 4.3 MIXED-PRECISION CHANNEL SELECTION

To validate the effectiveness of our four-strategy mixed protection, we conduct an ablation study on Llama3-8B and Qwen2.5-7B, as shown in Table 3. For each mixed-quantization configuration selected during the search, we apply single-strategy mixed quantization to the same set of weights ("Part Mixed") and, for comparison, apply it indiscriminately to all weights ("All Mixed"). The last row reports the performance of AutoMixQ under the same average bit-width, indicating that:

- AutoMixQ adaptively selects appropriate strategies for different weights and consistently outperforms any single-strategy approach. This confirms that flexible strategy selection leads to a more effective global mixed-quantization scheme.

- In some cases, considering both top- and low-magnitude channels yields better results than focusing solely on the top ones, confirming our earlier observation. Under sub-4-bit quantization, it is insufficient to protect only high-magnitude channels; certain low-magnitude but important channels must also be preserved.

### 4.4 VISUALIZATION OF THE SEARCH PROCESS AND RESULTS

We perform an ablation study on the distillation (KD) loss, with results shown in Figure 4. As illustrated, while the KD loss has little effect in the early stages of search, its benefit becomes increasingly clear as the bit-width decreases. At the same average bit-width, incorporating KD loss enables the search to find mixed-quantization schemes with higher accuracy.

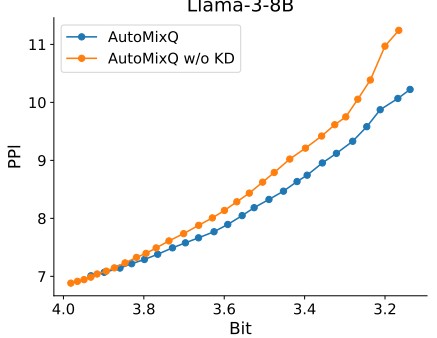

Figure 4: Search w/o knowledge distllation.

Table 4: Results of search and performance.

|  | Llama2-13B | Qwen2.5-7B |
|---|---|---|
| Search Time (GPU hours) | 88 | 84 |
| Compression Ratio (%) | 12.0 | 12.8 |
| Perplexity Degration (%) | 6.03 | 12.5 |

Table 3: Ablation study of quantization protection mechanism

| Activation | | Weight | | Part Mixed | | All Mixed 3.5-bit | | All Mixed 3-bit | |
|---|---|---|---|---|---|---|---|---|---|
| Top&Low | Top | Top&Low | Top | Llama | Qwen | Llama | Qwen | Llama | Qwen |
| ✓ | ✗ | ✗ | ✗ | 6.75 | 7.19 | 8.23 | 8.11 | 4.19e4 | 10.00 |
| ✗ | ✓ | ✗ | ✗ | 6.91 | 7.21 | 8.36 | 8.07 | 3.61e4 | 9.90 |
| ✗ | ✗ | ✓ | ✗ | 6.79 | **7.18** | 8.42 | 8.15 | 22.86 | 9.75 |
| ✗ | ✗ | ✗ | ✓ | 6.94 | 7.35 | 19.76 | 9.20 | 102.93 | 11.49 |
| ✓ | ✓ | ✓ | ✓ | **6.71** | **7.18** | **8.11** | **7.98** | **11.26** | **9.65** |

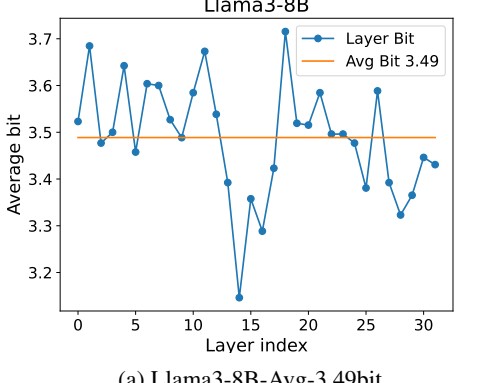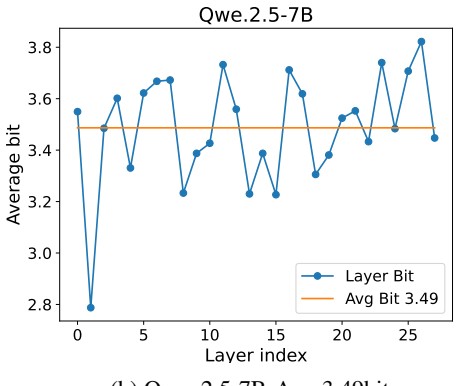

(a) Llama3-8B-Avg-3.49bit  (b) Qwen2.5-7B-Avg-3.49bit

Figure 5: The layer-wise bit allocation of search results.

Additionally, Table 4 reports the required search time and corresponding compression ratios for different models, together with perplexity degradation when AutoMixQ reduces precision from 4-bit to 3.5-bit. Figure 5 further illustrates the layer-wise and per-weight bit allocations produced by AutoMixQ. We observe distinct allocation patterns across models: for Llama, lower bit-widths are concentrated in the middle and later layers, consistent with previous studies (Yu et al., 2024; Daniel Han & team, 2025); in contrast, Qwen distributes bit-widths more uniformly across layers, except for the second layer.

## 5 CONCLUSION AND LIMITATIONS

In this paper, we present **AutoMixQ**, an automatic mixed-precision quantization framework tailored for sub-4-bit LLM deployment. Our approach introduces a mixed-precision protection strategy that considers both salient and inconspicuous channels: instead of protecting only large-magnitude outliers, it allocates higher bit-widths to channels with the largest and smallest absolute magnitudes based on quantization sensitivity, thereby preserving semantically important low-magnitude information. Furthermore, AutoMixQ integrates this protection with an automatic global search that inherently models inter-layer dependencies, enabling globally balanced bit allocation. Extensive experiments demonstrate that AutoMixQ achieves superior accuracy–efficiency trade-offs in the sub-4-bit regime, consistently outperforming existing PTQ methods.

Despite these gains, AutoMixQ still requires an average precision of about 3.5 bits to match AWQ int3 performance, leaving room for improvement toward a true 3.0-bit regime with stronger acceleration benefits. Moreover, current experiments are limited to pseudo-quantization, and real quantization with hardware-level validation remains future work. Nevertheless, we argue that AutoMixQ is a meaningful step forward, as it demonstrates the feasibility of sub-4-bit mixed quantization and offers the potential for acceleration once supported by future hardware.

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

# A APPENDIX

## A.1 AUTOMIXQ BASED ON 3-BIT PRECISION

Table 5: Perplexity evaluation (↓) on wikitext2 based on 3-bit.

|  | Llama | | Qwen | |
|---|---|---|---|---|
|  | 2-13B | 3-8B | 2.5-7B | 2.5-14B |
| GPTQ W3A16 | 5.36 | 10.06 | 8.21 | 7.01 |
| AWQ W3A16 | 5.33 | 8.14 | 8.18 | 6.64 |
| AutoMixQ W3A16 | **5.30** | **8.07** | **7.99** | **6.61** |

Table 6: Perplexity evaluation (↑) on zero-shot task based on 3-bit.

|  |  | MMLU | Hella. | PiQA | ARC_E | ARC_C | Wino. | Avg. |
|---|---|---|---|---|---|---|---|---|
| Llama2-13B | AWQ W3A16 | 48.67 | **58.47** | 77.15 | 77.82 | 45.82 | **72.61** | 63.42 |
|  | AutoMixQ W3A16 | **48.70** | 58.30 | **78.02** | **77.86** | **46.25** | 71.9 | **63.51** |
| Llama3-8B | AWQ W3A16 | 53.72 | 54.85 | **78.51** | 76.22 | 43.00 | 70.48 | 62.80 |
|  | AutoMixQ W3A16 | **54.38** | **55.37** | 78.24 | **76.56** | 43.00 | 70.48 | 63.00 |

To validate the effectiveness of AutoMixQ, we additionally report search results based on 3-bit quantization. We start searching from 3-bit quantization, and replace 3-bit with a mixture configuration of 2-bit and 4-bit. We provide the evaluation results on wikitext2 and zero-shot tasks, show in Table 5 and Table 6.

## A.2 VISUALIZATION OF WEIGHTS BIT ALLOCATIONS

We visualize detailed linear weight bit allocation produced by AutoMixQ on Llama3-8B, shown in Figure 6. In each figure, we plot the overall average bit-width as a reference. From the figures, we find that AutoMixQ tends to assign lower bit-widths to the attention modules (particularly Q, K, and V) during the search process, while allocating higher bit-widths to the MLP layers. We hypothesize that this phenomenon may be related to the sparsity level of the weights, and its quantization rationale behind AutoMixQ needs further exploration.

## A.3 THE USE OF LARGE LANGUAGE MODELS (LLMS)

We complete this paper with limited assistance of LLMs. Specifically, after finishing our manuscript, we use it to polish some words or sentences in some certain places.

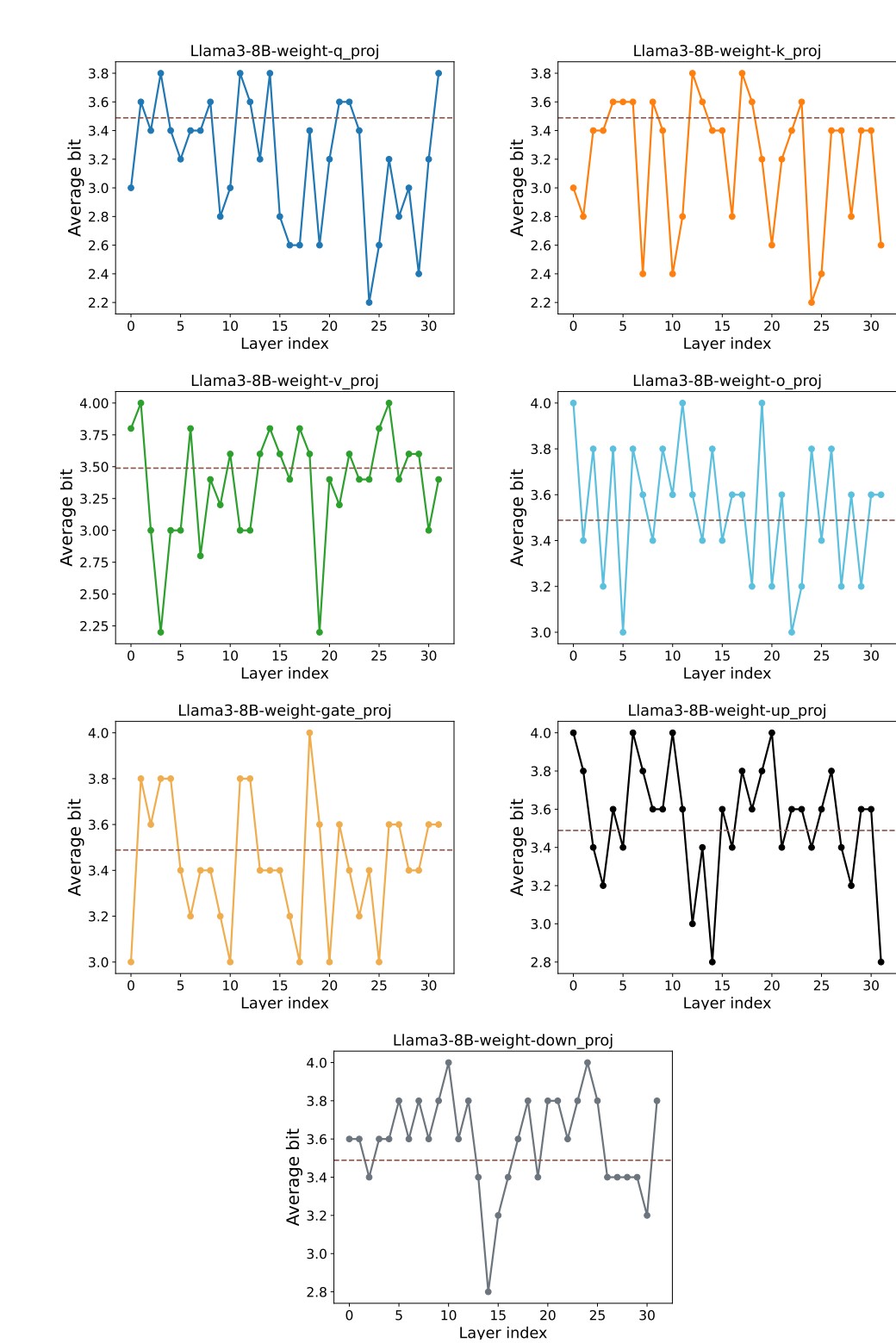

Figure 6: The different linear weight bit allocation of search results.

