# OpenReview forum: "AutoMixQ: Automatic Mixed-precision Quantization for Deploying Bit-Efficient LLMs"
_ICLR.cc/2026/Conference — ICLR 2026 Conference Withdrawn Submission_

### Official Review · Reviewer_wHi5 · 2025-10-28

**Soundness:** 1
**Presentation:** 1
**Contribution:** 2
**Rating:** 2
**Confidence:** 4

**Summary:**

The authors proposed a mixed precision post-training quantization strategy motivated by challenges of layerwise quantization strategies of prior work

**Strengths:**

- Proposed approach of looking at global instead of layer wise properties for quantization is interesting
- The argument that some weights may need higher bit width than others is compelling
- Approach tackles the challenging sub-4-bit regime
- Strong results of the proposed approach on downstream tasks
- Ablation studies are included

**Weaknesses:**

W0 Perplexity results of the proposed approach are worse than the GTPQ baseline (specifically, W3.75A16 with the proposed AutoMixQ is worse than W4A16 GPTQ)

W1 The method is not training free: a cross-entropy + distillation loss is used. Details remain unclear.

W2 Experiments seem incomplete
- Basic quantization methods have been excluded from the experimental results
- GPTQ is present in the perplexity results but not in downstream results.

W3 Presentation issues:
- The introduction claims that QAT approaches require full re-training. This is not the case. QAT can be very well be performed only in continual pre-training.
- Same sentence, it is mentioned that QAT requires labeled data, which is not true.
- The introduction claims that DeepSeek MoE would require 1200GB of GPU memory, yet the point of the MoE architecture is that only parts for the active parameters need to be loaded at a time.
- Alg 1: "mixde" -> "mixed"
- L306: "Paramerters" -> "Parameters"
- Best methods could be highlighted in Table 1

W4: Supposed gains do not translate to compute (hardware lottery) – but this is acknowledged.

**Questions:**

Q1 Regarding W1 (above), you describe that this composite loss function guides the procedure. However, in the algorithm the loss calculation is not mentioned. Could you please clarify?

Q2 If you do train during the quantization procedure, wouldn't it be fairer to consider QAT baselines? Does the AWQ baseline also use some training signal including a teacher model?

Q3: Why was GPTQ excluded from downstream analysis?

---

### Official Review · Reviewer_uAte · 2025-10-31

**Soundness:** 2
**Presentation:** 2
**Contribution:** 2
**Rating:** 2
**Confidence:** 5

**Summary:**

This paper proposes AutoMixQ, an automatic MPQ framework for sub-4-bit LLM deployment that addresses two claimed limitations of existing PTQ methods: channel-level oversight and layer-level myopia.
Experiments on Llama and Qwen models demonstrate improvements over GPTQ and AWQ baselines in sub-4-bit regimes.

**Strengths:**

**1. Straightforward framework.** The overall pipeline is simple and could serve as a baseline for future studies on mixed-precision PTQ.

**2. Visualizations of layer-wise bit allocation patterns.** Figure 5 provides layer-wise bit allocation visualizations across different models, revealing distinct allocation patterns that differ between model architectures.
This analysis offers some interpretability into how the method distributes precision across layers.

**3. Ablation studies included.** The authors conduct ablation studies examining the four protection strategies and the contribution of knowledge distillation loss, demonstrating some effort to understand component contributions..

**Weaknesses:**

**1. Insufficient comparison with state-of-the-art mixed-precision quantization methods.** The paper only compares against GPTQ and AWQ with naive random mixed-precision allocation, ignoring recent LLM mixed-precision quantization works such as SpQR [1], ResQ [2], and MixLLM [3].
The reviewer strongly encourages the authors to refer to the recent surveys including [4] for a more comprehensive understanding of mixed-precision quantization strategies.
Naturally, properly designed MPQ should yield performance improvements over uniform-precision counterparts.
That is a given - beyond that, it should also surpass prior state-of-the-art MPQ methods to justify its contribution.

**2. Weak methodological novelty and naive approach.** The proposed method is overly simplistic, consisting of (1) four heuristic strategies for selecting channels based on magnitude statistics, and (2) a brute-force perturbation-based search with standard distillation loss.
Neither component presents significant technical innovation; magnitude-based channel selection is well-established, and the search procedure is a straightforward random sampling approach without principled optimization.

**3. Poor presentation quality with numerous errors.** The paper suffers from multiple presentation issues including typos (lines 105, 369, etc.), poorly formatted tables and figures without proper captions or explanations (e.g., Figure ?? in line 195), missing experimental details, and inconsistent notation.
Tables 1-3 present results without sufficient context about what configurations were actually tested, making reproducibility questionable.
From the reviewer’s viewpoint, the writing could also be further improved.
Although the idea itself is simple and easy to understand, the writing lacks clarity and makes it unnecessarily difficult for the reviewer to follow the core message.

**4. Limited experimental evaluation.** The experiments are severely insufficient—only 4 models tested, no comparison with recent mixed-precision methods, missing critical analyses (e.g., search budget analysis, sensitivity to hyperparameters α and T, comparison of different search strategies) and no actual hardware deployment validation despite claiming "practical acceleration" benefits.

**5. Missing implementation and reproducibility.** No code implementation is provided, making it impossible to verify the described search algorithm, mixed-precision assignment process, or loss computation.
Given the simplicity of the proposed method, open-sourcing the implementation should have been trivial and would significantly strengthen the paper’s credibility and reproducibility.

**Questions:**

Refer to Weaknesses

**References**

[1] T. Dettmers et al., “SpQR: A Sparse-Quantized Representation for Near-Lossless LLM Weight Compression”, ICLR 2024

[2] U. Saxena et al., “ResQ: Mixed-Precision Quantization of Large Language Models with Low-Rank Residuals”, ICML 2025

[3] X. Wang et al., “MixLLM: Dynamic Routing in Mixed Large Language Models”, NAACL 2025

[4] M. Rakka et al., “Mixed-Precision Quantization for Language Models: Techniques and Prospects”, https://arxiv.org/abs/2510.16805v1

---

### Official Review · Reviewer_KsJw · 2025-10-31

**Soundness:** 2
**Presentation:** 1
**Contribution:** 2
**Rating:** 2
**Confidence:** 4

**Summary:**

This paper proposes AutoMixQ, a framework that automatically determines mixed-precision configurations for LLMs. During the search process, AutoMixQ enhances the mixed-precision model by considering not only salient channels but also those that are less salient yet may carry important semantic information as candidates for higher precision.

**Strengths:**

* Mixed-precision quantization for LLMs is an important and timely problem.

* The observation that relying solely on saliency for precision assignment may be suboptimal is interesting.

**Weaknesses:**

1. Methodology seems too heuristic.

The observation that considering only salient channels may be suboptimal is interesting. However, the proposed solution appears largely heuristic. According to my understanding, the method randomly selects a subset of non-salient channels along with salient ones for high-bit allocation. Given the authors’ claim that “saliency is not all we need,” I expected some principled insight or criterion addressing what else is important beyond saliency then. Unfortunately, the paper does not provide such an explanation or insight, leaving the approach appearing ad hoc.

* Missing important figure.

In line 195, there is a broken reference (“Figure ??”). At first, I assumed this was a minor typo, but I could not locate any figure that appears to correspond to it. This missing figure seems crucial, as it likely provides the key supporting evidence for the paper’s central motivation that non-salient channels can also have significant impact. Without it, the paper currently lacks quantitative or visual support for this motivation.

* Vague explanation of quantization protection mechanism.

The description of the four proposed strategies is unclear. For example, Strategy 1 assigns high bit-width to channels corresponding to both high and low output activations. However, in Figure 2, channels 2, 3, 5, and 6 appear to fall into these categories, yet only channels 2 and 3 are ultimately selected. It is unclear why channels 5 and 6 are excluded, or what mechanism determines this selection. Additionally, the meaning of the white columns (i.e., those that are neither highlighted as high nor low) is not explained. Do they represent 'medium'-magnitude activations? Are they entirely excluded from high-bit allocation, and if so, why? A clearer explanation of these mechanisms, ideally with concrete examples, is necessary for understanding.

* Ambiguity about AWQ evaluations.

To my knowledge, AWQ is a uniform quantization scheme. It is unclear how the authors evaluated 3.5- and 3.75-bit AWQ.

**Questions:**

Please see weakness section.

---

### Note · Authors · 2025-11-14

I have read and agree with the venue's withdrawal policy on behalf of myself and my co-authors.